# Bacitracin Supplementation as a Growth Promoter Down-Regulates Innate and Adaptive Cytokines in Broilers' Intestines

Gabriela C. Dal Pont [1,*], Annah Lee [1], Cristiano Bortoluzzi [1], Yuhua Z. Farnell [1], Christos Gougoulias [2] and Michael H. Kogut [3,*]

1 Department of Poultry Science, Texas A&M Agrilife Research, Texas A&M University, College Station, TX 77843, USA; yuhua.farnell@ag.tamu.edu (Y.Z.F.)
2 Innovad NV/SA, Postbaan 69, 2910 Essen, Belgium; c.gougoulias@innovad-global.com
3 USDA-ARS, Southern Plains Agricultural Research Center, College Station, TX 77845, USA
* Correspondence: gabrieladp@tamu.edu (G.C.D.P.); mike.kogut@usda.gov (M.H.K.); Tel.: +1-979-260-3772 (M.H.K.)

**Abstract:** In the past decade, the withdrawal of antibiotics used as growth promoters (AGP) has increased some poultry industry challenges, such as the rise of intestinal diseases. Experts advocate that AGPs improve performance due to the modulation of the intestinal microbiota, with resulting anti-inflammatory effects. However, the impact and interactions of AGPs with the host intestinal immune system are still unknown, which represents issues in developing effective alternatives for AGPs. Therefore, this study was aimed at better understanding the potential mechanism of action of bacitracin used as AGP and its impacts on the intestinal immune system. Ninety day-of-hatch chickens were randomly assigned to two treatments with three repetitions of fifteen birds, a control (CNT) group with a corn/soybean meal standard diet, and a control diet supplemented with 50 g/ton of feed of bacitracin (BMD). The cytokines' and chemokines' production (IFN-α, IFN-γ, IL-16, IL-10, IL-21, IL-6, M-CSF, MIP-3α, MIP-1β, VEGF and CCL-5) were assessed in the jejunum and ileum at 14, 21, 28 and 36 days of age by using a chicken-specific cytokine/chemokine peptide ELISA array. Broilers with BMD supplementation were found to have a lower production of IL-16, IFN-γ, M-CSF, IL-21, MIP-1β and VEGF in the jejunum at 14 d. However, from 21 through 36 days, the effect of BMD on cytokine production in the jejunum was negligible except for CCL-5, which was reduced at D36. In the ileum, BMD effects on the cytokine profile started at 28 d, when BMD-supplemented broilers showed a reduced IL-6 production level. At day 36, BMD reduced IL-16 and MIP-3α production but increased VEGF concentration in the ileum tissue. The present study demonstrated that the use of bacitracin as an AGP modulates the small intestine immune system, especially in the first phase of the broiler's life (up to 14 days). Moreover, BMD anti-inflammatory effects include not only innate immunity but also seemed to influence the development of the adaptive immune response as seen by the decreased production of IL-21 and IL-16. These results demonstrate that a commonly used AGP in broiler feed had a local anti-inflammatory effect.

**Keywords:** bacitracin; antibiotic; cytokine; interleukin; immunity

## 1. Introduction

The removal of antibiotic growth promoters (AGP) and the dramatic increase in antibiotic-free poultry production worldwide have resulted in increased biosecurity concerns, especially due to the rise of intestinal inflammation and pathologies as clinical and subclinical clostridial-induced necrotic enteritis [1]. This situation has brought awareness to intestinal health and the pursuit of alternatives to antibiotics and reduced intestinal inflammation. Different hypotheses have been proposed to explain the mode of action of AGP, such as the control of microbiota, the reduction of subclinical intestinal infections [2]

and anti-inflammatory effects [3]. Bacitracin methylene disalicylate (BMD) is one of the most widely used AGPs to improve productivity by the US poultry industry. The inclusion of BMD in chicken feed can decrease the number of *Clostridium perfringens* [4], benefit the gut wall morphology [5] and influence the immune system and immune responses [6]. While the literature on the impact of feed AGPs on performance and microbiota is extensive, few papers bring a comprehensive evaluation of the interactions of BMD with the host intestinal immune system.

The intestine is considered the largest immune organ of the body [7], and its inflammatory process is complex, involving intestinal cells (enterocytes), immune cells present in the organ and immune cells that migrate to the site of infection [8]. Cytokines are (glyco)peptide molecules with regulatory effects on hematopoietic, immune and tissue cells, and they are the main managers of the immune response. The cytokine group includes interferons (IFNs), interleukins (ILs), colony-stimulating factors (CSFs), transforming growth factors (TGFs) and tumor necrosis factors (TNFs) [9]. Classically, ILs have been described by their effects in lymphocytes and IFNs as antiviral, and the roles of CSFs, TGFs and TNFs can be predicted by their names [9]. Chemokines, a specialized group of immune cell chemoattractants, primarily regulate cellular traffic from the circulation. With the progress of research, we know that one cytokine can have multiple downstream effects. Beyond the complexity of the immune response, avian researchers face an additional barrier: the scarcity of immune-physiology studies in birds. The scarcity of research related to avian species most often results in the use of mammalian references for interpretations of avian cytokines. However, with the advance of technology and the availability of cytokine kits specific to avian species, researchers have the tools to perform a more comprehensive evaluation of the chicken intestinal immune response.

Therefore, we hypothesized that BMD as an AGP had direct anti-inflammatory effects on the intestine immune response. The current work aimed to test this hypothesis and evaluate how bacitracin supplementation might regulate the I See Inside (IIS) response. In this study, we quantified the production of several avian cytokines and chemokines in the intestinal tissue (jejunum and ileum) of broilers fed BMD in growth promoter concentration.

## 2. Materials and Methods

The experiment was conducted in accordance with guidelines for animal care set by the United States Department of Agriculture Animal Care and Use Committee (USDA IACUC #2019012). The trial was conducted at the Agricultural Research Service Facility of the United States Department of Agriculture (ARS-USDA), College Station, TX, USA.

A total of 90 Cobb male by-product day-of-hatch chickens were randomly assigned to two experimental treatment groups with three repetitions of fifteen birds (total of forty-five animals/treatment). The experimental groups were: (1) control (CNT) with a corn/soybean meal standard diet; and (2) control diet supplemented with bacitracin at a growth promoter dosage (BMD).

The animals were raised in floor pens with new shaving for up to 36 d of age. The diet furnished was formulated to meet or exceed broilers requirements (Cobb manual, 2018), and two feeding phases were used: starter (1–21 days) and grower (21–36 days). Water and feed were offered ad libitum and environmental conditions were maintained for each growing phase according to the Cobb recommendations.

## 3. Results

### 3.1. Necropsy and Sampling

Two birds from each pen (a total of six birds per treatment) were randomly selected for sample collection on days 14, 21, 28 and 36 of age. During the necropsy, samples of jejunum and ileum were collected for cytokine quantification. Intestinal anatomy was used to ensure the sampling of a similar segment in all birds, and the sampling was made in the middle of each segment. The intestinal sections were snap-frozen in liquid nitrogen and stored at $-80\ ^\circ\text{C}$.

### 3.2. Cytokine Panel Analysis

Concentrations of cytokines and chemokines were assessed in the jejunum and ileum. Intestinal tissue was lysed using BeadBug™ tubes (#Z763799-50EA, Sigma-Aldrich®, St. Louis, MO, USA) filled with lysis buffer solution (100 mg of tissue/1 mL of buffer) and homogenized with the Omni International Bead Ruptor Elite (Kennesaw, GA, USA). The lysis buffer solution was formulated with a 1:2 cell lysis buffer supplemented with protease inhibitor cocktail as recommended by the company (#AA-LYS and #AA-PI, respectively, RayBiotech Life Inc., Peachtree Corners, GA, USA). Then, the lysate was centrifuged for 10 min at 3000 rpm, and the supernatant was collected and diluted 1:2 in distilled 1xPBS solution to produce the final solution. If necessary, the supernatant was stored at $-80$ °C, and on the day of the analysis the lysate samples were thawed at room temperature and diluted in PBS.

The MILLIPLEX® Chicken Cytokine/Chemokine Panel was used to measure interferon-alpha (IFN-$\alpha$), interferon-gamma (IFN-$\gamma$), interleukins 16, 10, 21 and 6 (IL-16, IL-10, IL-21, IL-6), macrophage-colony stimulating factor (M-CSF), macrophage inflammatory protein 3 alpha (MIP-3$\alpha$), macrophage inflammatory protein 1 beta (MIP-1β), vascular endothelial growth factor (VEGF) and CC-chemokine ligand 5 (CCL-5, also called Regulated upon Activation Normal T cell Expressed and presumably Secreted, RANTES) (#GCYT1-16K, EMD Millipore Corporation, Billerica, MA, USA) read in the MagPix® System (Luminex Corporation, Austin, TX, USA).

### 3.3. Statistical Analysis

Data were analyzed according to a complete randomized design. At first, the normality of the data was verified through a Shapiro–Wilk's test. Data with normal distribution were analyzed using a *t*-test. Variables with non-normal distribution were analyzed with Kruskal–Wallis. The software JMP® Pro 15.0.0 (SAS Institute Inc., San Francisco, CA, USA) was used, and for all tests a *p*-value < 0.05 was considered statistically significant and $0.05 \leq p\text{-value} \leq 0.07$ considered tendency.

## 4. Results and Discussion

BMD supplementation in the feed influenced cytokine production, mainly at the jejunum at an early-stage post-hatch, i.e., D14. At this stage, BMD-fed broilers showed a lower concentration of IL-16, IFN-$\gamma$, M-CSF, IL-21, MIP-1β and VEGF in the jejunum than the control (Figure 1). From day 14 through day 36, the effect of BMD on cytokine production in the jejunum was negligible; the only exception observed was the reduction of CCL-5 at 36 d. In the ileum, the changes in the cytokine profile due to BMD supplementation appeared later than in the jejunum, with the initial changes appearing on day 28 post hatch, when reduced IL-6 production was observed. On day 36, bacitracin reduced IL-16 and MIP-3$\alpha$ production but increased VEGF concentration in the ileum tissue (Figure 2).

The results from this study confirm the hypothesis that AGPs have anti-inflammatory effects (3), as we demonstrated that BMD reduced the production of several pro-inflammatory cytokines in the small intestine. However, this effect was dependent on both age and location (intestinal segment). IL-16, IFN-$\gamma$, M-CSF, IL-21, MIP-1β, VEGF and CCL-5 (tendency) were found in lower concentrations in the jejunum in broilers fed with BMD by 14 days of age. Thereafter, the major influence of BMD was observed in the ileum, with a reduction in IL-6 at day 28, and in IL-16 and MIP-3$\alpha$ at day 36 post-hatch. The literature has reported that broilers fed a diet supplemented with BMD had a reduction in the gene expression (mRNA levels) of pro-inflammatory cytokines in the jejunum [6,10] and the cecum [11]. Therefore, our work corroborates and strengthens these findings by measuring the cytokines' protein production in the intestinal tissue instead of mRNA expression. Moreover, the data from this experiment indicate that there were fewer immune changes in the ileum driven by BMD than the jejunum and that the immune changes in the ileum appeared at a later stage, i.e., 28 d onward.

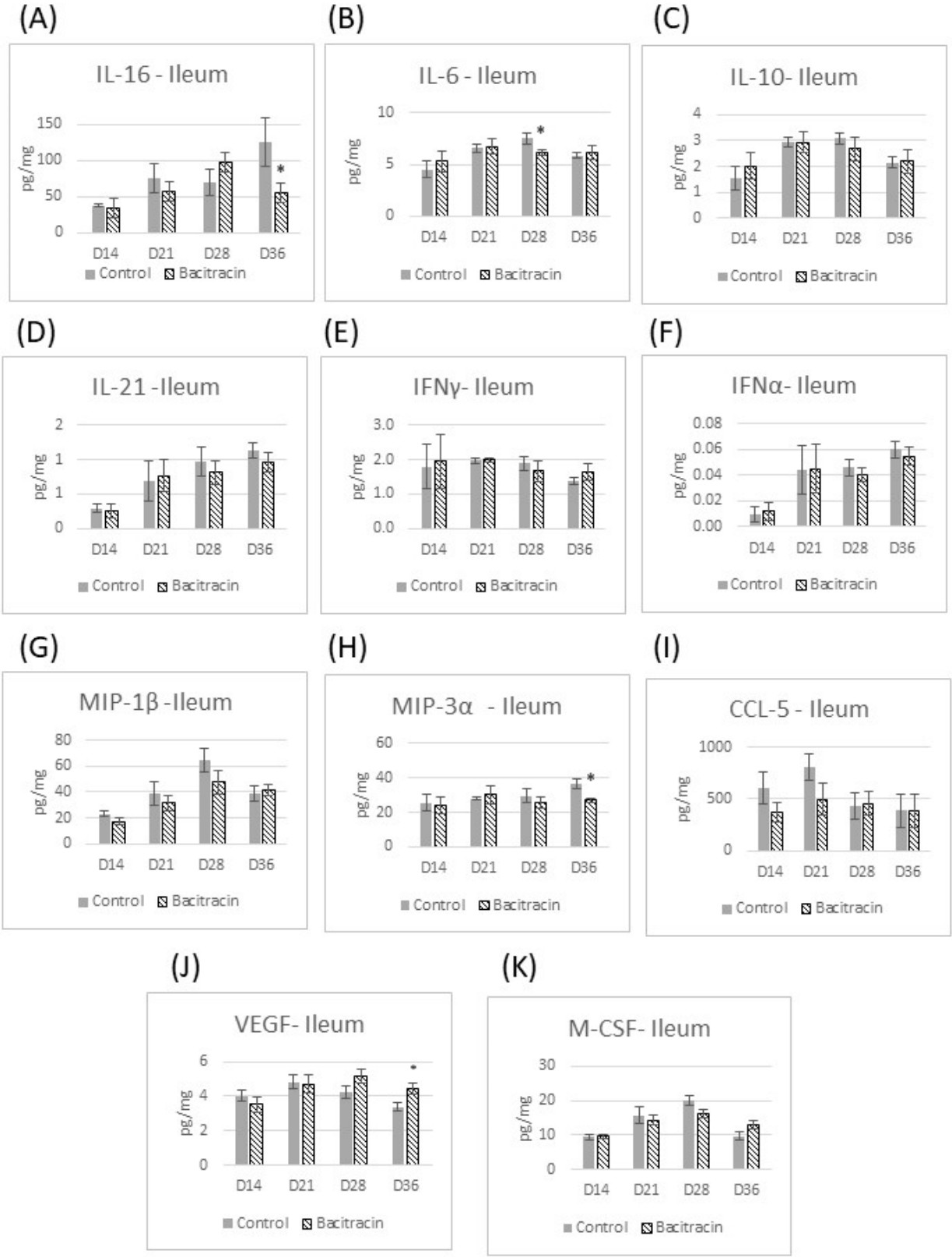

**Figure 1.** Concentration of cytokine or chemokines (pg of cytokine/mg of tissue) in the ileum of broilers fed with a regular corn–soybean based diet or control diet supplemented with bacitracin in a growth promoter dosage on days 14, 21, 28 and 36 of age; concentrations of: (**A**) IL-16; (**B**) IL-6; (**C**) IL-10; (**D**) IL-21; (**E**) IFN-γ; (**F**) IFNα; (**G**) MIP-1β; (**H**) MIP-3α; (**I**) RANTES; (**J**) VEGF; and (**K**) M-CSF. * Statistical difference with $0.001 < p\text{-value} < 0.05$. $n = 6$ animals/treatment.

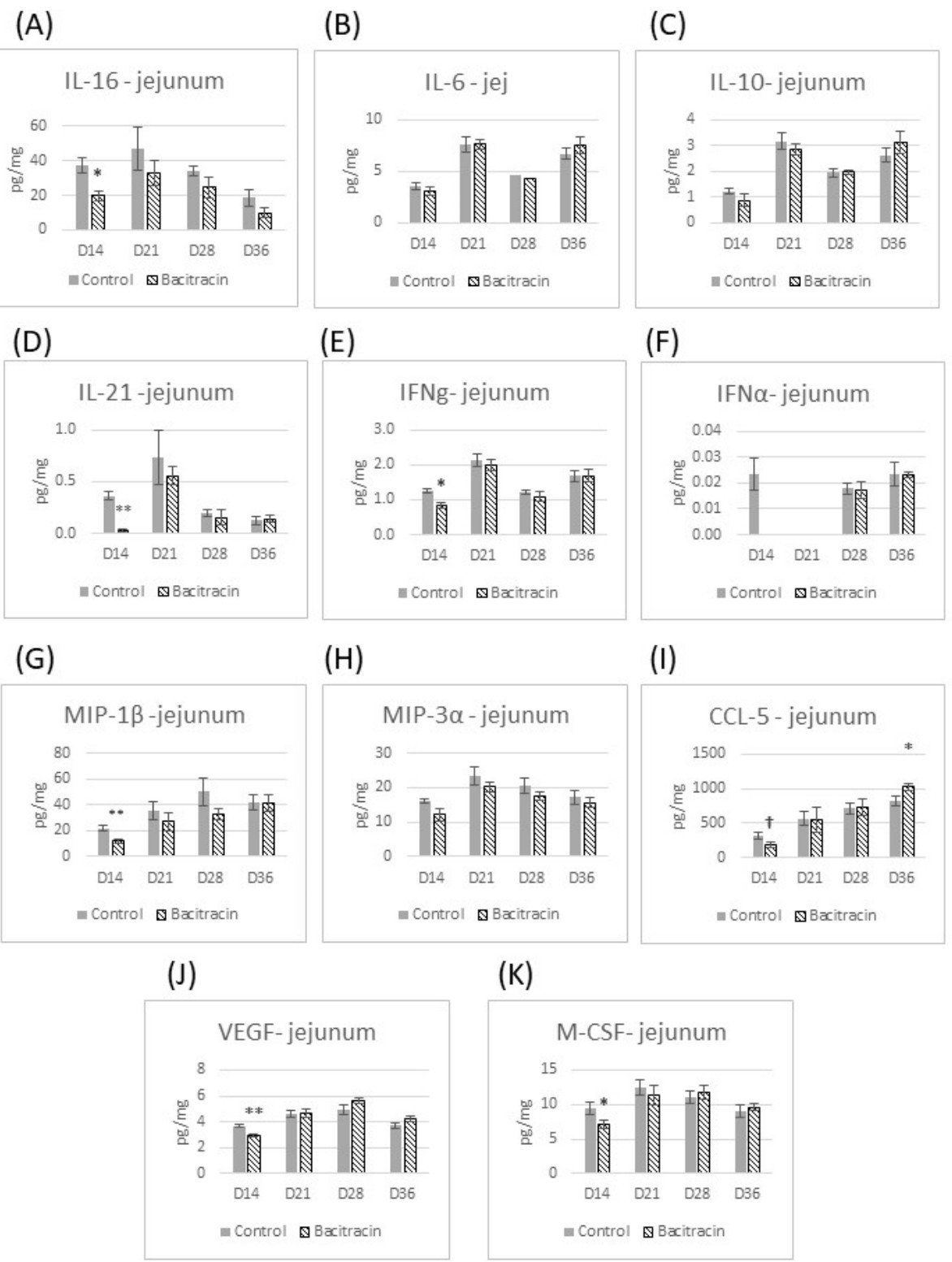

**Figure 2.** Concentration of cytokine or chemokines (pg of cytokine/mg of tissue) in the jejunum of broilers fed with a regular corn–soybean-based diet or control diet supplemented with bacitracin in a growth promoter dosage on days 14, 21, 28 and 36 of age; Concentrations of: (**A**) IL-16; (**B**) IL-6; (**C**) IL-10; (**D**) IL-21; (**E**) IFN-γ; (**F**) IFNα; (**G**) MIP-1β; (**H**) MIP-3α; (**I**) CCL-5, also known as RANTES; (**J**) VEGF; and (**K**) M-CSF. * Statistical difference with $0.001 < p$-value $< 0.05$; ** Statistical difference with $p$-value $< 0.001$. † Tendency ($0.05 \leq p$-value $\leq 0.07$). $n = 6$ animals/treatment.

M-CSF and MIP-1β (also known as CCL-4), macrophage stimulatory cytokines, and IFN-γ, IL-16, IL-21, VEGF and CCL-5 are all produced by T cells. The reduction of these cytokines with BMD supplementation indicates the multiple effects of BMD on T lymphocytes, and indirectly indicates the importance of T lymphocytes for the homeostasis of the intestine, and perhaps the phenotype of broilers fed BMD. The effects of BMD on the adaptive immunity in broilers have been described previously. For example, BMD reduces the mRNA levels of the adaptive cytokines IL-10, IL-4 and IFN-γ in the ceca of broilers at early stages of growth (7 and 14 d) [11], and pro-inflammatory T-cell cytokines (IL-17, IL-2 and IFN-γ) in the jejunum [10]. Thus, from 14 to 36 days post hatch, BMD acts by regulating the cytokine response of innate and adaptive immune cells, thus reducing the potential environmentally induced and low-grade chronic-intestinal inflammation, which can negatively affect the performance of broilers. Moreover, BMD feed supplementation can modify the intestinal microbiota of chickens (2), even though its effects are subtle compared to other AGPs [12,13]. Therefore, we cannot exclude the possibility that, in the present study, BMD modulated chicken microbiota and the different microbiota and the derived metabolites had anti-inflammatory effects on the chicken intestine, this being an important topic for future investigation.

IL-16 is a chemoattractant for CD4+ T cells and monocytes, and inhibits the activation of Th2 cells [14,15]. Studies in mammals indicate that IL-16 is upregulated in intestinal inflammation disorders and is involved in the mechanisms of Inflammatory Bowel Disease [16]. In the present work, BMD supplementation reduced the IL-16 concentration in the broiler's small intestine. If IL-16 is, indeed, involved in chicken intestinal inflammation, as in other species, its reduction by BMD may explain one of BMD's intestinal anti-inflammatory effects and the consequent enhancement in intestinal morphology present in the literature with BMD use. Additionally, the reduction in intestinal inflammation might collaborate for performance improvement since the energy and nutrients used for the immune response would be directed to muscle production.

The present experiment is the proof of concept for the use of bacitracin as an AGP modulating primarily the intestinal innate immune system, especially in the first phase of the broiler's life (up to 14 days). However, the down-regulation of the innate immunity by BMD also affects the adaptive immune response. Lastly, the current work contributed to the existing knowledge by measuring specific chicken immuno-protein concentrations directly in the intestinal tissue, which gives more confidence for the assumption of immune responses than the mRNA gene expression of the same targets.

**Author Contributions:** Conceptualization, G.C.D.P., C.G. and M.H.K.; methodology, G.C.D.P., A.L., C.B. and Y.Z.F.; software, G.C.D.P. and M.H.K.; validation, G.C.D.P., A.L., C.B., Y.Z.F., C.G. and M.H.K.; formal analysis, G.C.D.P. and M.H.K.; investigation, G.C.D.P., A.L. and C.B.; resources, G.C.D.P.; data curation, G.C.D.P., A.L. and C.B.; writing—original draft preparation, G.C.D.P.; writing—review and editing, G.C.D.P., A.L., C.B., C.G. and M.H.K.; visualization, G.C.D.P.; supervision, Y.Z.F. and M.H.K.; project administration, M.H.K. All authors have read and agreed to the published version of the manuscript.

**Funding:** This research received no external funding.

**Institutional Review Board Statement:** Ethics Committee Name: United States Department of Agriculture Animal Care and Use Committee. Approval Code: USDA IACUC #2019012. Approval Date: February 2019.

**Informed Consent Statement:** Not applicable.

**Data Availability Statement:** Not applicable.

**Acknowledgments:** Mention of commercial or proprietary products in this paper does not constitute an endorsement of these products by the USDA, nor does it imply the recommendation of products by the USDA to the exclusion of similar products.

**Conflicts of Interest:** The authors declare no conflict of interest.

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
