# Peer review of "Bacitracin Supplementation as a Growth Promoter Down-Regulates Innate and Adaptive Cytokines in Broilers’ Intestines"

_poultry, doi:10.3390/poultry2030030_

Round 1

Reviewer 1 Report

The work reported here in this 'short communication' is of great value. It's a very well written manuscript and the findings are very useful and support the conclusion. I only have two minor comments. Pl see attached.

Author Response

Response 1: changed to 'also seemed to influence'

Response 2: thank you, the main concern was for statistical analysisof the data.

Reviewer 2 Report

The topic this short communication covers is important, however, the quality of the communication is poorly prepared, and it is very clear that not much effort was put in preparing the communication even though it is clear that a lot of effort was put on the testing and data collection parts. Several mistakes could have been easily avoided if this was read prior to submission. Details on edits are in the attached file. 

One major point needs to be reviewed and discussed is the effect of diet phase change on immune response, the communication does not show the formulations offered to birds, however, the fact that the feed phase was changed has an effect on immune response and that part should be addressed in this communication.

no comments

Author Response

The authors have agreed with all of the editorial comments made by the reviewer and have revised the manuscript accordingly.  

Reviewer 3 Report

1. Please notify that 6 birds per treatment were tested for those factors in Abstract. 

2. The conclusion, "bacitracin modulate small intestine immune system", is too general. Cytokines and chemokines could not present the whole immune reaction or system happens in the GI tract. Please revise it in Abtract and the conclusion section.

3. Remove lines from 86 to 89 which belongs to the MDPI format. And provide necessary information from lines 197 to 253.

4. Figure 1 title said ... in jejunum but the figures said ...ileum. The same with Figure 2. I also suggest converting the figure to table. A figure with a-f section is too complicated.

5.what is the "IIS" in line 69?

6. Using "g" instead of "rpm" in line 105.

Author Response

The authors agreed with all of the editorial comments made by the reviewer and have revised the manuscript accordingly.